# Using Regulatory Flexibility to Address Market Informality in Seed Systems: A Global Study

**Katrin Kuhlmann** [1,*] **and Bhramar Dey** [2]

1   New Markets Lab, Georgetown University Law Center, 600 New Jersey Ave., N.W., Washington, DC 20001, USA
2   Catholic Relief Services, 228 W Lexington Street, Baltimore, MD 21201, USA; bhramar.dey@crs.org
*   Correspondence: kak84@georgetown.edu

**Abstract:** Seed rules and regulations determine who can produce and sell seeds, which varieties will be available in the market, the quality of seed for sale, and where seed can be bought and sold. The legal and regulatory environment for seed impacts all stakeholders, including those in the informal sector, through shaping who can participate in the market and the quality and diversity of seed available. This paper addresses a gap in the current literature regarding the role of law and regulation in linking the informal and formal seed sectors and creating more inclusive and better governed seed systems. Drawing upon insights from the literature, global case studies, key expert consultations, and a methodology on the design and implementation of law and regulation, we present a framework that evaluates how regulatory flexibility can be built into seed systems to address farmers' needs and engage stakeholders of all sizes. Our study focuses on two key dimensions: extending market frontiers and liberalizing seed quality control mechanisms. We find that flexible regulatory approaches and practices play a central role in building bridges between formal and informal seed systems, guaranteeing quality seed in the market, and encouraging market entry for high-quality traditional and farmer-preferred varieties.

**Keywords:** seed systems; seed policy; seed regulations; agricultural law; enabling environment

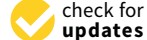



## 1. Introduction

Inclusive seed systems that deliver high-quality seed of new and improved varieties across a range of crops can be catalytic for agricultural transformation, improvements in productivity, and food security. In order to be inclusive, a seed system should allow for stakeholders of all sizes to participate in the market, including both commercial companies and small farmers, and should promote both commercially attractive crops (e.g., hybrid maize) and other crops desired by farmers and local communities, including open pollinated and vegetatively propagated crops.

Seed markets and the regulatory enabling environment function as interconnected systems. For purposes of this study, the term "seed systems" is used to refer to the totality of processes that are part of development, maintenance, production, storage, and diffusion of cultivars [1]. The term "enabling environment" refers to the multi-layered system of policies, laws, regulations, and other measures that govern seed systems and other aspects of agricultural development. These different regulatory processes are interconnected components of the seed value chain, and regulatory interventions at one stage of the chain have an effect on other stages [2,3].

Within seed systems, scholars and experts often make an important distinction between "formal seed systems" and "informal seed systems," even though these systems exist in parallel. Formal seed systems tend to include both government regulation and public institutions that govern private industries engaged in scientific plant breeding [4]. In many countries, the formal seed sector centers around registered and certified seed, mainly of hybrid crops, and encompasses activities along the seed value chain carried

out by seed companies, distributors, breeders, importers, and exporters [5]. "Informal" or "farmer" seed systems include small-scale farmers who usually produce and exchange their own varieties (primarily open pollinated or vegetatively propagated varieties), often for non-commercial purposes [6]. These systems are viewed as largely outside of the system of formal regulation, although they are dependent upon farmers' knowledge [7] and are often subject to effective forms of "informal regulation." However, as this study highlights, formal and informal systems often coexist, explicitly and implicitly, both in practice and within legal and regulatory frameworks. In a survey conducted by FAO, in 25 percent of countries, commercial production and exchange of uncertified seeds was permitted for a select group of crops, while 29 percent of countries explicitly banned the sale of seeds that were not certified [8]. Ideally, a country's policy, legal, and regulatory framework should address the needs of both the formal and informal sectors [4,9]. In many countries, informal seed sector participation is considerably higher than formal seed sector participation, with eighty to ninety percent of food grains dependent upon the informal sector [10,11]. In countries such as Kenya, Zimbabwe, and Malawi, the informal sector supplies ninety percent of seed sourced to smallholders [12]. In these markets, new varieties are largely obtained from informal sourcing, and many of the transactions in the market are informal [12]. While many of these transactions take place outside of formal legal channels, the legal and regulatory framework does shape informal market activity. As our study explores, the structure of rules and regulations can either link the formal and informal sectors or further cause separation and generate inefficiencies within the market.

Accordingly, we evaluate the interplay between formal and informal systems within the agricultural sector in terms of regulatory design and, to the extent possible, implementation.

The interface between formal and informal seed systems plays a major role in the introduction, dissemination, and scaling of seed production of new and improved varieties that benefit smallholders at the last mile. Two dynamics are particularly important in considering the connection between informal and formal seed systems and the role of law and regulation at their intersection.

First, farmers need increased choices of high-quality crop seeds and planting materials to diversify their portfolios and improve livelihoods. This diversity at the last mile is essential for farmers to absorb and adapt to shocks, thus increasing resiliency. Often, many of those crop seeds are not available in the market or are not produced by seed companies, because they may not be profitable for their businesses. In such cases, farmers have sourced seeds from neighbors or other farmers or have used their own saved seeds and or bought "potential planting materials" from the market. Potential planting materials refers to the high-quality well sorted grains that farmers purchase from grain traders and local sellers, with research organizations and private seed companies as the main producers of new climate-smart and disease-resistant varieties. National and international agricultural research centers do produce varieties that are climate-smart and biotic/abiotic-stress-tolerant. However, in order to get these improved varieties to farmers, the legal and regulatory environment needs to be designed to ensure that a wide range of improved varieties can be release and disseminated in the market.

Second, farmers should be able to determine how they wish to operate in the market. For some, informal, trust-based systems may be preferable to more heavily regulated, formal markets. However, in most markets, the barriers to entry to formal markets can be so high that farmers are effectively left with little choice. Additionally, capacity building and adoption of alternative seed quality assurance mechanisms (allowed under national regulations) could help to ensure that informal seed system actors have flexibility in formalizing their roles and actions [13,14]. Again, the design and implementation of laws and regulations can play a direct role in farmers' ability to choose their own fate.

This paper reviews what we call "regulatory flexibilities" [15], which address both dynamics and can be a catalyst for increased farmer choice and agency. We have identified these flexibilities based on an existing methodology and prior work as well as interviews with seed system experts and illustrative country case studies. Our goal is to highlight

examples of regulatory flexibilities around two key thematic study areas that connect the informal and formal seed sectors: expanding market frontiers so that farmers have enhanced choices as well assuring the quality of seeds and planting materials. In both thematic areas, a set of key findings emerge both at regulation of market entry (ex-ante regulation) and at the enforcement stage (ex-post regulation) [16].

- Theme One—Extending Seed Market Frontiers: Regulatory systems often dictate who can sell what seed and in which markets. These aspects are typically governed through measures such as registration of seed dealers and functions, which sometimes include venues. Governments may also restrict the sale of seed to registered entities selling formally certified seed, leaving smaller farmers outside of recognized market channels. These regulatory aspects of "seed market frontiers" will directly impact whether farmers can access seed of the right quality and variety at the right price to increase on-farm productivity. Flexibility in who can sell seed (seed producers, vendors, and dealers) and in which market locations can be particularly important for the informal seed sector. Innovations such as the seed clubs in Vietnam [17] that have enabled community-based seed schemes have been particularly helpful in extending seed market frontiers for farmers and have strengthened quality control through local government participation and support as well. In addition, flexibility in the rules and guidelines governing seed variety registration and release and plant variety protection (PVP) and plant breeders' rights (PBR) will have a longer-term impact on access, availability, and affordability of seed.

- Theme Two–Liberalizing Seed Quality Control Mechanisms: Reliable seed quality is a shared concern across seed systems, and many regulatory systems are designed to monitor the quality of seeds before they reach the market. Different approaches have emerged to balance policymakers' interest in guaranteeing the quality of seed in the market while encouraging market entry for high quality traditional varieties. These can include systems that blend formal seed certification with more flexible models (including quality declared seed (QDS), self-certification, and truth-in-labeling approaches) and those in which the certification process has been fully or partly privatized (including authorization of private seed inspectors).

This review neither champions increased regulation for informal seed actors nor argues for less regulation for formal seed actors. Instead, the goal of our work is to highlight an array of examples, drawn mainly from legal and regulatory systems of countries in the global south, so that national and regional policymakers can make informed decisions around whether to introduce regulatory flexibilities for seed systems that could increase choices for farmers at the last mile.

The methodology and study design are presented in Section 2. Sections 3 and 4 discuss the thematic areas of the study and relevant findings. Finally, concluding remarks are provided in Section 5.

## 2. Materials and Methods

The study was based on desktop research and comparative legal and regulatory analysis, along with semi-structured interviews with key industry and global seed policy experts based on a set of questions on the thematic study areas. The study's approach was also based on a methodology developed by Katrin Kuhlmann and the New Markets Lab to identify "regulatory flexibilities" and innovation that enable governments to address the needs of different stakeholders, including small farmers. Regulatory flexibility may take the form of different rules for different types of business, rules designed to address the needs of a certain group (for example, in response to discrimination or economic vulnerability), or some other differentiation [15] that appears in the context of regulatory design or implementation. Regulatory design can include (1) how a legal and regulatory system is oriented in terms of market activity (i.e., do regulatory measures focus on actions before market activity takes place (ex-ante approach) or enforcement once market entry has occurred (ex-post approach)) [16]; (2) the purpose of rules and regulations, i.e.,

whether measures are tailored to address a particular objective, including, for example, inclusive market development, food security, or sustainable development [15]; and (3) the degree of regulatory flexibility, i.e., whether regulatory approaches are designed to adapt to different stakeholders' needs, diverse realities, and other considerations [15]. Regulatory implementation aspects include the degree to which rules and regulations make systems more efficient; regulatory preconditions or gateways that act as hurdles to market participation (including the extent to which one regulatory process hinges upon completion of another); inclusive rulemaking that engages diverse stakeholders; and effectiveness of rules and regulations [15].

Throughout this study, regulatory flexibility and innovation are highlighted in the context of the seed value chain [2] (see Figure 1), incorporating, in particular, the ways in which the enabling environment addresses the needs of the many small farmers who make up informal seed systems. The labeling, packaging (including small packets), marketing, and distribution stage also encompasses seed dissemination. Regional harmonization of seed rules, which is particularly comprehensive in sub-Saharan Africa, will impact much of the regulatory value chain, in particular variety registration and release, seed certification, and trade. For the purposes of this study, we have assessed regulatory flexibility based on the text of laws and regulations, relevant analysis and literature, discussions with key stakeholders and global experts, and select case studies. The global scope of the study allowed us to compare the legal and seed regulatory systems of a set of countries to draw comparisons between systems. Focus countries were determined based on informant interviews, literature review, and the authors' past experience. We found that few countries show no flexibility and that several countries show a range of regulatory flexibilities across both study areas. The following sections elaborate upon this approach in the context of the two thematic study areas.

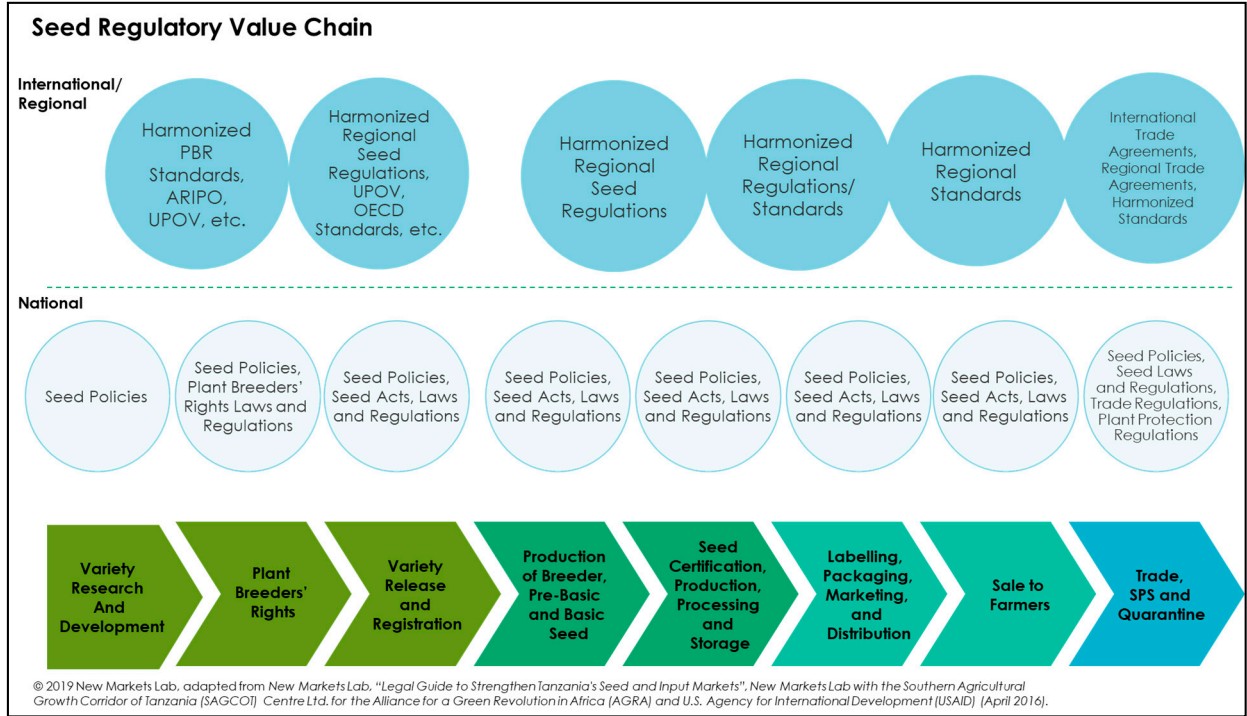

**Figure 1.** Seed regulatory value chain (adapted from [2]). Key: Plant Breeders' Rights (PBR); African Regional Intellectual Property Organization (ARIPO); International Union for the Protection of New Varieties of Plants (UPOV); Organisation for Economic Co-operation and Development (OECD); and Sanitary and Phytosanitary Measures (SPS).).

In order to illustrate the connection between the regulatory elements that appear at different stages of the value chains and farmers' needs, we adapted the methodology

and regulatory value chain framework discussed above to align common regulatory elements relevant to seed systems with farmers' abilities and needs, drawing upon categories elaborated in work by Bert Visser [9]. This overlay of common regulatory elements with farmers' needs is depicted in Figure 2. Regulatory elements are shown horizontally and color-coded, with the second part of the figure mapping these regulatory elements against farmers' abilities to select, breed, and register varieties; acquire the seeds of their choice; save, exchange, and use farm-saved seed; establish farmers' enterprises and groups; and sell and commercialize varieties. This allows us to visually depict our evaluation of the role of the regulatory system in influencing or shaping farmers' needs.

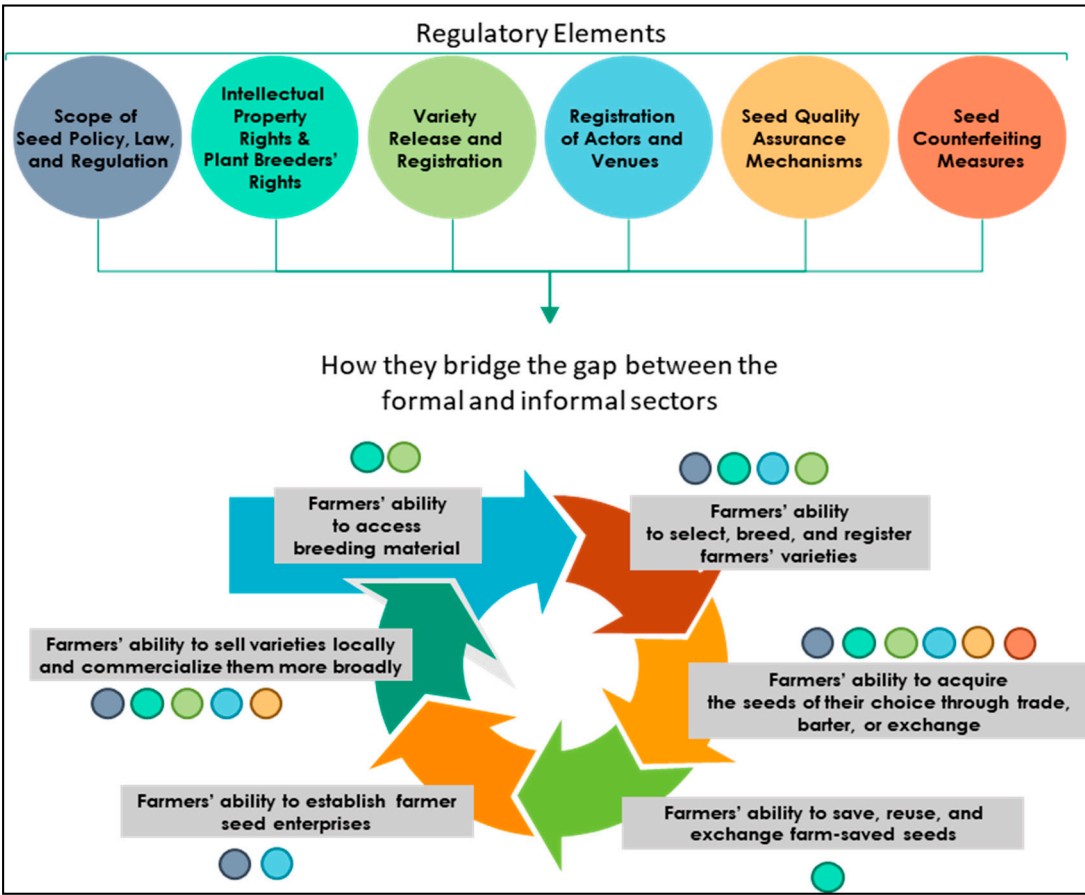

**Figure 2.** Relationship between regulatory elements and link between informal and formal seed sectors (adapted from [2,9]).

This study focuses on regulatory design, mainly because regulatory implementation would require extensive field-based analysis to evaluate some of the models discussed in the paper. However, several aspects of regulatory implementation are also worth noting, particularly efficiency of rules and regulations; regulatory preconditions or "regulatory gateways" (the extent to which one regulatory process hinges upon completion of another); inclusive rulemaking; and effectiveness of rules and regulations [15]. Although implementation is largely outside of the scope of this study, we highlight examples of regulatory inclusivity and effectiveness wherever possible, drawing upon key case studies.

## 3. Results—Thematic Area: Extending Seed Market Frontiers

### 3.1. Regulatory Aspects of Extending Seed Market Frontiers

By design, national seed policies and regulations do not always treat all seed the same. Some policies and regulations are broad in scope and cover every crop and activity in the seed value chain, while others apply to only certified seeds or specific crops. Within these systems, rules and regulations may be tailored by design (de jure) or implementation

(de facto) to focus on commercially attractive crop varieties, such as maize, excluding traditional varieties that are an essential part of the traditional knowledge and farming practices of local communities.

Many countries and regions—including the European Union, Switzerland, and several African countries—reflect a strong focus on commercial crops in their policy and regulatory systems [8,18]. In many cases, these systems do not include explicit exemptions for farmers and the informal sector [6,9].

On the other hand, some seed regulatory systems are more "differentiated," meaning that they incorporate some elements of formal seed regulatory systems but recognize differences among actors, type of activities, or certain regulatory functions [15]. In Kenya, for example, the informal sector is not subject to all the legal and regulatory requirements by which the formal sector is governed. This allows the informal sector to operate alongside the formal sector while still ensuring that the seed sold and labeled as higher-quality seed meets strict requirements [19]. India's system exhibits a differentiated approach in several regulatory areas, and Vietnam's seed clubs could also be considered a differentiated approach.

While not many countries have established a fully differentiated approach, our analysis shows that quite a few contain important regulatory flexibilities that address farmers' needs and realities. These flexibilities can include regulatory exceptions, tailored legal and regulatory requirements, and different procedures for smaller farmers, as explained below.

Although the scope and scale of seed regulations vary across countries, many countries' systems dictate who can legally produce and sell seed for specific markets, an aspect we refer to as "market frontiers." In many markets, seed must be certified by the government before they can be sold by registered seed dealers at registered points of sale, such as a business premises/agro-dealer shops. However, in more remote areas, last-mile informal shops and kiosks are important points of sale in the informal seed sector. In addition to the rules governing registration of seed actors and venues, regulatory flexibility should be considered regarding the registration of crop varieties and plant variety protection (PVP)/plant breeders' rights (PBR) [20]. Figure 3 depicts the regulatory considerations related to thematic area one, expanding market frontiers, within both the formal and informal seed sectors and benchmarked against farmers' needs and abilities. Within this thematic area, several regulatory elements along the value chain are relevant to addressing farmers' needs and abilities, namely the scope of policies, laws, and regulations; intellectual property and plant breeders' rights rules; regulations on variety registration and release; and rules on registration of actors and venues. Because so many regulatory elements impact market frontiers, regulatory flexibilities are particularly important, as they can directly impact farmers' ability to access breeding material; select, breed, and register their varieties; acquire and exchange seeds of their choice; save, reuse, and exchange seed; establish farmers' seed enterprises; and sell varieties locally and commercialize them more broadly. The regulatory flexibilities related to these needs and abilities are discussed in greater detail below, and several are worth highlighting. With respect to farmers' ability to save, reuse, and exchange farm-saved seed (as well as farmers' ability to breed and acquire desired varieties), flexibilities in crop variety registration and plant breeders' rights approaches, discussed below, are particularly relevant. This includes protections under Article 9 of the International Treaty on Plant Genetic Resources for Food and Agriculture (ITPGRFA) [21] Several countries show flexibility in variety registration and release, as discussed below, and others (e.g., Malaysia) have incorporated flexibility on plant breeders' rights into their legal systems. Ultimately, how farmers' genetic resources are accessed, bred, and perhaps released in the market entails a number of particular legal and regulatory considerations and deserves greater focus in the context of legal and regulatory systems. With regard to farmers' ability to establish farmer seed enterprises, Vietnam's seed clubs (see Box 2) are particularly noteworthy.

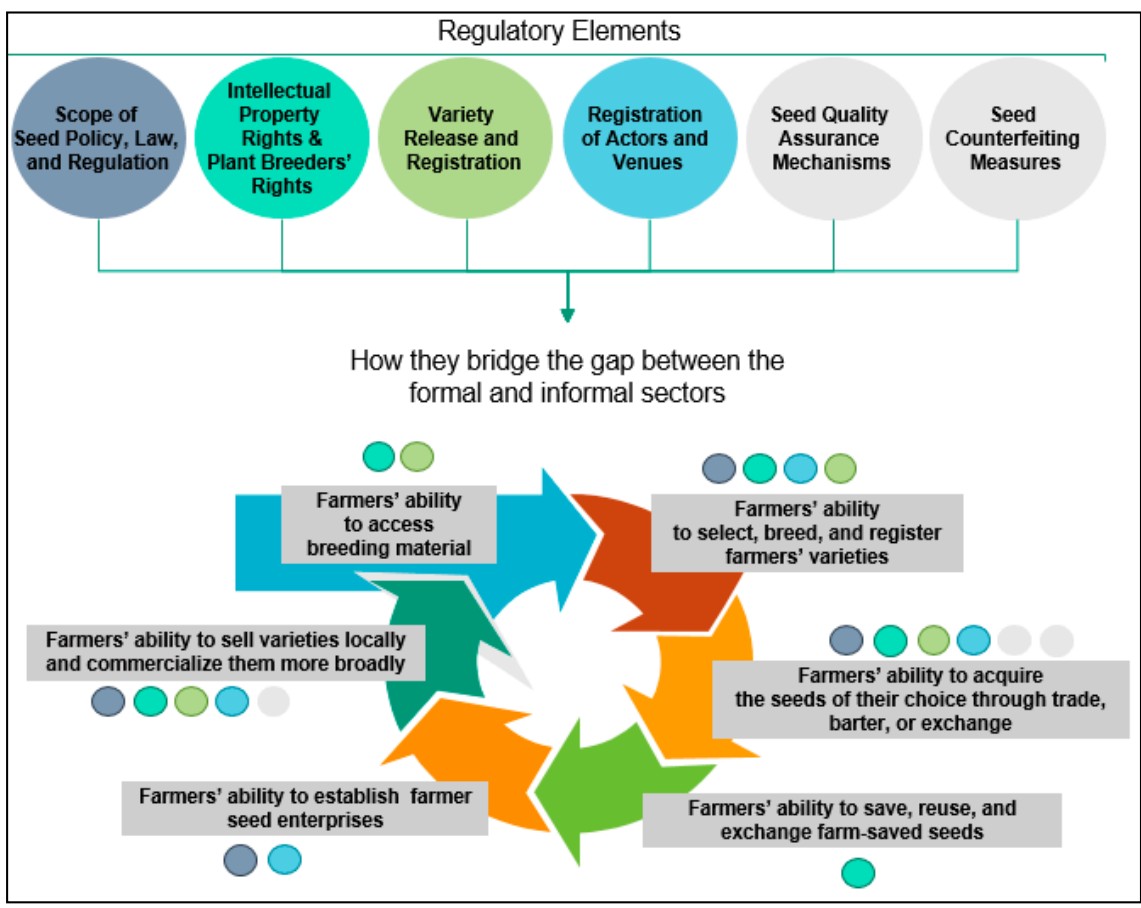

**Figure 3.** Regulatory elements of extending market frontiers (adapted from [2,9]).

### 3.2. Flexible Regulatory Approaches to Registration of Seed Actors and Venues

Regulatory factors play a central role in determining which stakeholders can engage in the market and which seeds can be introduced into the market, impacting the diversity of seed sold. As a common "regulatory gateway" [15], many countries place regulatory limits on actors and activities in the seed market, including who may sell what kind of seed where, which is typically monitored through registration or licensing requirements.

Registration of seed producers, distributors, or dealers is a form of ex-ante regulatory control to limit who participates in the seed market and under which conditions. Often, registration is required under the umbrella of "seed dealers," but it can also include registration of seed producers, farmers, breeders, and other actors. While less common, some countries also require registration of venues where seed can be sold.

Registration can fulfill a legitimate policy function, such as guaranteeing the quality of the seed introduced into the market by controlling and supervising the individuals and enterprises involved in the seed sector. However, when implemented in practice, these measures are often cumbersome and expensive, effectively raising barriers to extended market participation. Requiring registration of every actor involved in seed systems can, however, tip the balance towards formal seed systems and can also present difficulties for regulators, particularly when combined with weak institutional capacity.

Some countries' regulatory systems go so far as requiring a specific education level to become registered as a seed dealer. Others require seed dealers to have a registered business with a physical address and perhaps even a certain annual business income level. For example, in Peru, to obtain a license to commercialize seed, seed dealers must show proof of a license for a commercial venue, location, and contact information for the vendor [22].

Regulatory approaches regarding the registration of seed actors range from strict registration requirements, under which every actor involved in the exchange of seed must be registered, to more flexible approaches with exceptions for informal actors, to differentiated requirements for the informal and formal sectors (see Table 1). Strict or comprehensive registration falls on one end of the spectrum and is typically required to guarantee that seed sold is of the highest quality. However, implementation of comprehensive registration measures requires institutional resources that are not always available. It also imposes a higher burden on the private sector, which can impact access and availability of seed in the market and consequently affect market diversification and expansion. One example of this approach is Colombia, which, through the Instituto Colombiano Agropecuario (ICA), requires registration of every person or organization involved in the production, import, export, or storage of seeds. Colombia's regulations also prohibit small-scale farmers from sharing seeds or storing or commercializing other farmers' seeds if they are not registered or certified without ICA's authorization [23].

More flexible regulatory approaches also exist that either differentiate explicitly between the informal and formal sectors or establish exemptions for certain actors or activities (e.g., small-scale farmers, activities with non-commercial purposes, etc.). India is an example of a differentiated system (see Box 1), where formal and informal actors are explicitly recognized and regulated separately. Kenya also recognizes the formal and informal sectors as separate and distinct, but Kenya does limit the venues where certain types of seed (non-commercial varieties) can be sold [19]. Other countries do not draw such explicit distinctions between the informal and formal sectors but provide flexibility and some exemptions for certain actors and activities. Table 1 summarizes the different regulatory approaches to registration of seed actors and venues. Overall, both differentiated registration approaches and partially flexible approaches can improve access to seed systems while maintaining quality in the market.

**Box 1.** India's differentiated regulatory approach for registration of seed sector actors.

> India has one of the more flexible regulatory approaches for registration of seed system actors, which regulates formal and informal actors under different measures. Currently, the Indian Ministry of Agriculture and Farmers' Welfare regulates the production, distribution, and sale of varieties in the formal seed sector, while the informal sector is regulated under the National Seeds Policy, 2002. In India, any person selling, importing, or exporting seed is referred to as a seed dealer and is mandated to comply with the guidelines in the Seeds Act (1966), Seeds Control Order (1983), and the New Policy on Seeds Development (1988), which collectively form the basis for the promotion and regulation of the seed industry. The Seeds (Control) Order 1983 establishes the registration procedure and requires any person involved in selling, storing, importing, or exporting seeds to obtain a license to be a seed dealer, which is valid for three years unless suspended or cancelled. India's National Seed Policy exempts farmers from compulsory registration and allows them to produce and sell seed varieties freely. Under India's new Seeds Bill 2019 [24], farmers will be able to sell seed on their own premises or in the market, as long as seed is not sold under a brand name. Farmer's rights to continue using the varieties of their choice is not limited by compulsory registration.

India's system is particularly noteworthy (Box 1), due to its high degree of flexibility and largely separate regulation of the formal and informal sectors. Vietnam also has a differentiated approach, and a 2008 decision of the Ministry of Agriculture and Rural Development formally recognized the informal seed sector (mainly through seed clubs; Box 2). These differentiated approaches provide more space for informal actors but often do place some limitations on the type of seed they can sell and where such seed can be sold. Overall, these approaches are examples of a regulatory design approach that can open the doors for informal seed actors.

**Table 1.** Regulatory approaches to registration of seed actors and venues (illustrative examples).

| Registration Approach | Country | Key Takeaways |
|---|---|---|
| **Comprehensive Registration** | **Colombia**<br>• Registration required for all actors of the seed regulatory value chain (Resolución 970, Instituto Colombiano Agropecuario). | Strict regulatory approaches can exclude smallholder farmers and other informal actors from the system. |
| **Differentiated Registration** | **India**<br>• Registration of formal and informal actors is regulated separately, and non-registered farmers can sell their own varieties freely (India National Seeds Policy, 2002).<br>• Under the New Seeds Bill, which is not yet enacted, farmers will be able to sell seed on their own premises or in the local market, provided that the seed is not branded (India New Seeds Bill, 2019).<br>**Vietnam**<br>• Registration is required, but the 2008 Ministry of Agriculture and Rural Development Decision (Decision 35/2008/QD-BNN) officially recognized the informal seed system (Quang et al., 2011).<br>• Smallholder farmers exchanging their own crops are exempt from registration but are required to ensure the quality of the variety and environmental sanitation (Vietnam Seed Ordinance and Regulation on Plant Protection and Quarantine, the Law on Environment Protection).<br>**Kenya** (Kenya's approach falls somewhere between differentiated approaches and the partially flexible approaches described below)<br>• Formal and informal actors are recognized, but the informal sale of seeds is limited to non-commercial varieties (Kenya National Seed Policy, 2010). | Differentiated regulatory approaches establish different requirements for formal and informal actors and often provide the most flexibility.<br>Flexible approaches can result in extended access to seed systems, while maintaining quality standards and registration controls.<br>Even when flexible, most systems impose some restrictions on venues or crop varieties that can be traded by informal actors. |
| **Partially Flexible Registration (Exemptions)** | **Peru**<br>• Seed researchers, producers, and traders are required to register.<br>• Actors involved in the production, trade, and storage of traditional varieties are exempt from registration (Seed Regulation 2012).<br>**Brazil**<br>• Distinction between commercial seed producers and smallholder farmers.<br>• Smallholder farmers are exempt from registration when exchanging seed amongst themselves (Brazil, Law No. 10.711 Ruling on National System for Seeds and Seedlings 2003).<br>**Tanzania**<br>• Registration is required for all actors in the seed regulatory value chain; however, flexibilities do exist for informal seed actors, with exchange of QDS allowed without registration (Tanzania Seed Regulations, 2007 (as amended 2017).<br>**Myanmar**<br>• Registration is required for all actors in the seed regulatory value chain.<br>• Exemptions are provided for "peasants" exchanging among themselves and small-scale farmers exchanging seed for non-commercial purposes (Myanmar 2015 Seeds Act). | Other systems exempt smallholder farmers from registration under certain circumstances, generally when trading with other informal actors.<br>While these do not draw as much of a distinction between formal and informal actors as the differentiated approaches, these exemptions can allow smallholder farmers and other informal actors to participate in seed systems. |

**Box 2.** Case study—Vietnam's seed clubs.

Organized seed groups, including "key seed suppliers" [9] or community-level seed enterprises [25], can be one way to offset the costs of regulatory requirements, such as individual dealer registration, and can also build trust surrounding seed sales in rural areas. These may include seed villages, seed clubs, or production centers that are trusted sellers of quality seed.

In Vietnam, seed clubs have played a strong role in the development of inclusive seed systems [9]. In 1996, the Southeast Asia Regional Initiatives for Community Empowerment (SEARICE) collaborated with international partners to develop the Community Biodiversity Development and Conservation (CBDC) program, which led to the formation of the seed clubs. Initially, farmers in Vietnam's seed clubs produced seed for their own use and for use within the local farming community; however, over time, farmers formed seed production groups and sometimes even produced varieties for formal registration and certification. In some cases, seed clubs have evolved into seed enterprises with the legal right to commercialize and sell seed [17]. In the An Giang Province, 212 seed clubs were established over 10 years (which met 90 percent of the province's demand for rice seed), and 28 evolved into seed companies with local government support [17].

The national government did play a critical role in the seed clubs' evolution as well, although it did so at a critical stage and then effectively transferred oversight to the provincial level. The seed clubs were recognized by the national government through a decision by the Ministry of Agriculture and Rural Development that enabled households and communities to engage in seed production and distribution (Decision 35/2008/QD-BNN) and approved the establishment and registration of the seed clubs, although governance was then effectively transferred to the sub-national level, and Village People Councils issued a decision allowing for the establishment and registration of seed clubs and establishing the right for farmers to carry out rice breeding activities [17]. Early on, the seed clubs' activities focused on plant varietal selection (PVS) and plant varietal rehabilitation (PVR), and they then shifted to participatory plant breeding (PPB) in 1997 after partners and organizations realized and accepted that farmers can and should also practice plant breeding [17].

Vietnam's seed clubs have been particularly successful in rice production in the Mekong Delta [26], beginning with HD1 rice [17]. As of 2018, seed clubs had produced 360 farmer-developed rice seed varieties, of which five had been certified as national seeds and four were undergoing testing for registration and certification [27]. Some seed club members became part of the provincial formal seed system through establishment of seed cooperatives, while others became part of the formal seed system by setting up small seed companies. Overall, Vietnam's seed clubs have helped smaller farmers overcome the costs associated with registration and secure market presence.

Other countries recognize the informal sector through partially flexible regulatory approaches that carve out certain enterprises or activities through regulatory exceptions. Brazil, for example [28], mandates that all persons, physical and juridical, who produce, improve, package, store, analyze, trade, import, and export varieties, be registered, except for smallholder farmers [29]. Brazil's Seed Law makes an exception for family farmers, agrarian reform settlers, and indigenous peoples who multiply seed or seedlings for distribution, exchange, or trade among themselves. This special treatment extends not only to actors who work with traditional, local, or Creole cultivars but also to registered varieties if they are traded and exchanged among family farmers [29]. However, this exemption is somewhat constrained under the seed regulations [30], due to an exclusion for family farmer organizations, which can only distribute, and not sell, seeds of local, traditional, or creole cultivars (farmers' groups are also subject to a separate regulatory process in order to appear in the National Register of Organizations (this applies to associations, cooperatives, and unions) [31].

Peru's system bases registration of seed producers on the type of varieties being produced and sold, with exceptions for traditional varieties (farmer's varieties). Even though Peru's General Regulation of the General Seeds Law requires registration of seed researchers, producers, and traders, it does not establish registration requirements for farmers as maintainers, seed producers, and sellers of traditional varieties [32]. Under a related regulatory flexibility in Peru's system, unregistered "non-certified seed" (native or local) may be commercialized if the seed producer or the field is registered or if the producer takes full responsibility for the quality of the seed [9]. These practices are intended to help streamline informal farmers' access to the seed market and promote the sustainable use of plant genetic resources and conservation of agro-biodiversity.

Tanzania's regulatory system also shows some flexibility for informal seed actors. Tanzania's seed regulations do require that any legal person or entity involved in the trade of seeds be registered as a seed dealer by the Tanzania Official Seed Certification Institute (TOSCI). However, informal seed actors are permitted to trade in some seeds, mainly QDS seeds, without being registered if they (i) produce the seed on their own farms; (ii) declare the quality of their own seed; and (iii) sell their seed to nearby farmers within an administrative area only [33]. In addition, Tanzania's Seed Regulations establish an appeal procedure before the Minister of Industry and Trade when applications for seed dealer registration are denied [2].

Myanmar is another example of a partially flexible approach. Myanmar's 2015 Seeds Act requires that any person involved in the production or importation of plant varieties for commercial purposes obtain a certificate [34], but it includes an exemption from registration for "peasants" who may distribute and sell varieties amongst each other [9]. Moreover, Myanmar's 2016 National Seeds Policy exempts small-scale farmers from registration when selling for non-commercial purposes to other small-scale farmers.

Myanmar's system is also notable because the Department of Agriculture (Seed Division and Extension Division) established a Seed Village Scheme under the National Seed Policy to organize community seed production [34]. Organized seed groups such as these are an important example of an approach that has built flexibility into heavily regulated (ex-ante) systems and enhanced farmers' ability to establish recognized seed enterprises.

Seed clubs have been particularly successful in Vietnam (see Box 2), and they have helped to overcome regulatory hurdles with respect to both market frontiers and seed quality control. Vietnam's seed regulations do require registration of seed sector actors, but smallholder farmers are exempt from registration requirements when they exchange their own crops, although they are required to ensure the quality of plant varieties and environmental sanitation. Seed clubs have helped farmers to address standards and enter markets, effectively acting as a seed guarantee system for market supply [17].

Vietnam's seed clubs are particularly noteworthy due to the balance between national and sub-national government oversight and support. As noted in Box 2, while the national government created the legal space for community seed production and distribution, the village councils established a process for registration of seed clubs and community plant breeding. Vietnam's seed clubs have also benefitted from support from provincial agriculture extension officers, seed centers, and research institutes. Some seed clubs' members became part of the provincial formal seed system through establishment of seed cooperatives, while others became part of the formal seed system by setting up small seed companies.

Overall, while the registration of seed actors is sometimes flagged as a bar to market entry, especially for smaller producers, there is considerable variation in countries' regulatory systems that has allowed for inclusive seed systems. Regulatory gaps do remain, however. As some expert consultations highlighted, greater regulatory oversight is needed over seed handlers, such as those involved in the transport and storage of seed. Currently, regulations tend to focus either on the development and production of seed varieties or on their sale. However, transport and storage of seed are delicate stages in the value chain, since the conditions under which seed is stored and transported can impact quality and germination. In addition to serving as a collective vehicle for seed dealer registration, organized seed groups or other approaches could apply in other areas as well, and transport and storage of seed could be an important area for future analysis. Myanmar's Seed Village Scheme and Vietnam's Seed Clubs have, in part, addressed this consideration through seed centers with appropriate infrastructure and technology for seed processing, storage, and marketing.

*3.3. Flexible Regulatory Approaches to Seed Variety Registration*

Another regulatory aspect that impacts market frontiers (and the diversity of varieties available in the market) is the regulatory control of which seed varieties enter the market,

which is also often done through a registration process. Many countries around the world require that crop varieties be registered and then formally released in the market, with national seed catalogues established to provide a formal inventory of available varieties. In such systems, before seed can be registered and released, it must usually be tested for Distinctness, Uniformity, and Stability (DUS) and Value for Cultivation and Use (VCU) or national performance trials. This process is another example of a "regulatory gateway" before seed can be commercialized and sold.

The formal regulatory process for variety registration and release can be burdensome for farmers and local communities due to regulatory complexity, potentially long timelines for variety release, and related economic investment. The testing and registration process may also not be suitable for all crop varieties. For example, VCU testing may not be appropriate for vegetable varieties, even though governments may deem it necessary for grain varieties. It is important to note that some countries do not mandate DUS or VCU testing (also known as national performance trials), and most commercial seed companies will conduct DUS testing independent of a government-managed testing process. Traditional or landrace varieties may have significant difficulty meeting regulatory requirements, since these varieties are naturally more variable (i.e., they do not always perform in a manner that is "uniform" and "stable" and hence would have difficulty passing the DUS test) [20].

A more flexible approach to variety registration would help to integrate the informal and formal sectors and empower smallholder farmers. It would also help to ensure the integrity and availability of varieties that are intentionally more variable in order to enhance resilience, even if these varieties do not meet the requirements of conventional DUS testing. Some countries' regulations do include flexibilities that could allow for the registration and sale of farmers' or landrace varieties [8]. Ethiopia is an example of a system with a highly flexible approach to variety registration, whereby smallholder farmers are exempt from variety registration when exchanging and selling farm-saved seed [35]. Vietnam's 2004 Seed Ordinance [36] does require DUS and VCU testing and multi-location trials, but farmers and farmers' organizations have the right to engage in rice breeding, selection, and production [17], and farmers' saved seed is covered under the 2008 Decision (see Box 2).

Flexibility may also be applied to the tests that are required for variety registration and release. Malaysia, for example, applies an "identifiability" approach to assess varieties based on a combination of qualities that are more heterogenous, making it more suitable for landraces [37]. The "identifiability test" essentially modifies the DUS test for varieties developed and bred by farmers, local communities, and indigenous populations by replacing the elements of uniformity and stability with identifiability, which is better tailored to the nature of these varieties.

Peru's General Seed Law includes clauses to promote the registration of native varieties "that can be exploited economically" by exempting these varieties from trial payments and taxes with the Register of Commercial Cultivars [9]. Peru also has a special regime for native potatoes, an integral component of Peruvian agriculture. Pursuant to Ministerial Resolution 0533-2008-AG, Peru's National Institute of Agricultural Innovation (INIA) implements, maintains, and updates the National Registry for Native Potatoes, adding new varieties of native potatoes and sharing useful technical data on native potatoes with farmers [38]. The registry promotes international and social recognition of native potato seed producers but does not grant breeder's rights for the registered varieties.

Several countries treat variety registration for landraces differently than that for commercial varieties and have introduced flexibility in their systems through establishment of alternative seed catalogues. Examples include Peru, France, Italy, the Netherlands, Costa Rica, Benin, Nepal, Finland, Switzerland, Republic of Korea, and Ecuador, all of which have all established alternative seed variety lists for registration of farmers' varieties [20].

In Brazil, there are two seed registries, one of which is the National Register of Local, Traditional, and Creole Varieties, which is managed and administered by the Ministry of Agriculture (MDA) [26]. Varieties in this registry must: (i) have been developed by family farmers or indigenous populations, (ii) possess distinctive characteristics for their

community, (iii) have been used in the community for more than three years, and (iv) not been developed by genetic engineering or industrial processes. The second registry is the National Cultivar Registry (NCR) for seed varieties produced in the formal seed sector, which must pass the DUS requirements. Brazil's system also allows farmers to save seeds from their yields for future production, although it imposes annual quantity limits. In Brazil, farmers' varieties in the National Register of Local, Traditional, and Creole Varieties are not eligible for commercial sale, and entities that register a seed variety ordinarily exchanged amongst local communities are prevented from applying for exclusive ownership rights (plant breeders' rights).

Benin also follows a flexible approach with different variety lists. Benin's national seed catalogue is comprised of three lists: List A includes varieties that need to undergo testing for both DUS and VCU in order to be registered and released, List B includes varieties that have been tested only for DUS and are eligible for exportation only, and List C includes landraces and new varieties developed through participatory plant breeding practices that only have to undergo only VCU tests in order to be registered and released [39]. Exempting landraces from DUS testing is a way to promote the inclusion of the informal seed sector, given that landraces, which are less uniform in nature and often fail to express the same characteristics when multiplied, tend to face challenges meeting DUS requirements [20].

These alternative approaches can allow farmers to sell and exchange materials without compliance with the strict regulatory requirements established for commercial varieties [20,40]. Some countries also use more flexible measures, such as guidelines (which are easier to change than laws and regulations) to update variety release and registration procedures, which can present another avenue for including local seed varieties and farmers' communities [20].

Expert consultations highlighted that even though many countries tend to regulate variety registration and release through a formal process at the national level, procedures at the sub-national level can sometimes be more flexible, as the Vietnam case study above highlights. For example, in Laos, national regulations prescribe rigorous requirements for the release of new varieties that effectively exclude more traditional crops. However, procedures at the provincial level tend to be more flexible and have allowed for the release of traditional varieties and their commercialization in the provinces [40].

Provincial rulemaking sometimes occurs because of national rules, but, in some cases, sub-national rules can build a case for changes at the national level. In addition, while the national and sub-national levels need to work collaboratively, some degree of autonomy between national and sub-national governments can also be important. Sub-national regulation does not work well in every country, however, and if there is little support from the national government to implement provincial procedures, the process can be risky in practice.

There are also links between regulatory processes at different stages of the value chain. Some of the experts consulted suggested that the QDS process (see Section 5 on quality control mechanisms) could provide insight into regulatory flexibilities that may be possible for crop variety registration. QDS seeds sometimes will not need to undergo formal variety registration in order to be sold in the market, but they are still subject to quality control measures.

There is a regional aspect to variety release and registration as well, and several regional economic communities (RECs) in Africa have developed rules on regional variety registration and regional variety catalogues, which are intended to allow for registered varieties to be freely traded within a region. Although some implementation challenges remain, most RECs are progressing with harmonization [41].

While concerns have been raised with the ability of regional systems to allow for more flexible regulatory approaches, including those discussed above, regional harmonization and recognition of informal seed systems need not be mutually exclusive. In the Southern African Development Community (SADC), for example, the SADC harmonized system provides for the registration of landrace varieties in the SADC Variety Catalogue. SADC's

system states that the procedures should identify characteristics that are essential for variety registration while considering the difficulties landrace varieties would face in applying DUS and VCU tests. Although procedures for this have not yet been established, experts consulted in the development of this study saw this as a positive development. SADC's system also makes a link with QDS in this regard (SADC is one of the only RECs to formally recognize QDS as a seed class), further supporting possible links between more flexible variety registration and QDS.

### 3.4. Flexible Regulatory Approaches to Plant Variety Protection

Legal and regulatory systems for intellectual property rights (IPR) for seed, including PVP and PBR systems, have implications for the development of inclusive seed markets. Although a full discussion of this issue is outside the scope of this paper, PVP and PBR can have an impact on extending market frontiers, since these systems are often related to which varieties enter the market and who can protect and control those varieties under what conditions. Under the ITPGRFA, farmers' varieties are also subject to negotiations for access and benefit sharing based on the Nagoya Protocol on Access and Benefit Sharing [42]; however, it is not clear whether international instruments could create an affirmative intellectual property right for farmers' varieties. PVP protection is also contingent on other regulatory processes, and varieties are required to be registered to be protected, which is an example of a "regulatory precondition or gateway" [15] that can exclude from protection varieties that do not meet registration requirements (this would include farmers' varieties, traditional varieties, etc.). Greater understanding of these considerations, and perhaps also identification of better approaches to address them through legal protections, could help to improve farmers' ability to acquire the seeds of their choice and save, exchange, and use farmer-saved seed. A system that incorporates other elements, such as connecting community seed banks with international and national seed banks, could help increase uptake for the development of more plant varieties [9,43].

More flexible approaches to PBR can be an important consideration in bridging the gap between the informal and formal sectors—see [20,44,45]—and our review is focused on a balance between IP and the rights of farming communities to extend the seed market in an inclusive manner to promote wider access to seed technology [46]. Examples of countries with flexible approaches in this area include India, Peru (Box 3), Thailand, Ethiopia, Malaysia, and Vietnam. It is also important to note that existing international regulatory frameworks leave some flexibility to countries in how to design regulatory approaches [9,47], but this flexibility often needs to be asserted by a country in order to be effective. The International Union for the Protection of New Varieties of Plants (UPOV) allows for "farmers' privilege", which enables farmers to save seeds from one season to another and use them, without authorization of the breeder, although exceptions apply to production, offering for sale, and marketing. Recent collaboration between the public and private sectors and UPOV regarding other aspects of farmers' privilege suggests that the privilege could be enhanced. One such discussion involves interpretation of "private and non-commercial use" in the context of "farmers' privilege," which could indicate a possible shift at the international level and corresponding flexibility in national law. The World Trade Organization (WTO) Agreement on Trade-Related Aspects of Intellectual Property (TRIPS Agreement) also gives countries some flexibility to design their national regulations through an "effective sui generis system" [48].

India's Protection of Plant Varieties and Farmers' Rights Act is broad in scope and applies to new plant varieties, extant (domestic and existing) varieties, and farmers' varieties and gives concurrent rights to breeders, researchers, and farmers. An apex body called the Protection of the Plant Varieties and Farmers' Rights Authority was established in India on 11 November 2001 to implement the Plant Varieties and Farmers' Rights Act.

**Box 3.** Case study: Peru's protection of traditional knowledge.

---

Peru has implemented reforms to incorporate protection for traditional knowledge into its IP system and make the system more flexible. Older literature on Peru's approach classified it as restrictive; however, recent reforms make it a more inclusive system. Peru currently regulates PBR through several legal instruments, including a law aligned with UPOV, Decision No. 345 Establishing the Common Regime on the Protection of the Rights of Breeders of New Plant Varieties, that allows farmers and farmers' organizations to apply for PVP, but varieties must still meet the novelty and DUS requirements. More recently, Peru also adopted the Comisión de Alto Nivel Anticorrupción (CAN) legal texts, which incorporate the protection of traditional knowledge, including agriculture, fisheries, health, horticulture, forestry, and environmental management.

---

Malaysia's system is notable as well, and the "identifiability test" referenced in relation to variety registration and release also allows farmers, local communities, and indigenous groups to seek PBR based on this tailored standard [37].

*3.5. Key Findings: Regulatory Approaches for Extending Market Frontiers*

Market frontiers tend to be regulated through ex-ante measures such as registration of enterprises, registration of crop varieties, and other measures that determine who can enter the market and what seed can be bought and sold. Navigating these regulatory requirements can be particularly challenging for the informal sector due to the time, cost, and complexity involved. Increasing regulatory flexibilities and reducing the number of regulatory gateways for informal seed actors can expand market frontiers and enable farmers to buy and sell the type of seed they want at the last mile. This can allow for expansion of the crop variety portfolio, leading to increased farmers' abilities and market systems that can absorb and adapt to shocks. Regulatory flexibility can also contribute to greater resiliency at the last-mile, which is essential during emergencies and pandemics such as COVID-19.

- Adopting flexible approaches for the registration of seed actors and crop varieties can allow for the inclusion of informal seed actors—and local or traditional seed varieties—in the market. Registration of seed sector actors is often a regulatory gateway to other activities, such as legally selling seed in the market, and rules and regulations in this area have a particularly significant impact on farmers.
- If countries do choose to require registration of seed sector actors, different regulatory options exist. These range from registration requirements that explicitly differentiate between informal and formal actors (India's system draws this distinction, for example) to more flexible approaches to registration with exemptions for small farmers selling certain types of seed (examples include Peru, Brazil, Myanmar, Tanzania, and Vietnam).
- Some countries also maintain flexible approaches to crop variety registration and release, including registration of farmers' varieties [20,40], as illustrated by the cases of Peru [38], Brazil, Benin, and other countries. These can include differentiated variety registration procedures with reduced testing requirements (Benin, for example, only requires VCU or national performance trials for landrace varieties). Flexible approaches can also include the adoption of different seed catalogues or variety lists for local, landrace, traditional, and farmers' varieties.
- For plant variety protection, some countries emphasize farmers' rights and may even allow protection for traditional varieties, as illustrated by the systems of India and Peru. This regulatory element could have an impact on farmers' ability to save, exchange, and reuse farmer-saved seed. International agreements, particularly the International Treaty for Plant Genetic Resources for Food and Agriculture, do allow some flexibility, but countries must exercise it and recognize rights for farmer-saved seed; see [9,15].
- Informal actors can also be integrated into the system through legal recognition of community and farmers' associations and seed clubs, such as the seed clubs that have

driven inclusive seed system development in Vietnam [17], seed village schemes in Myanmar, and seed cooperatives in Zimbabwe.

- As the Vietnam case study also highlights, sub-national government can play a critical role in registration of farmers and farmers' associations as well as registration of crop varieties, particularly when some autonomy exists vis-à-vis the national government. As has been the case in Vietnam, local government can also facilitate quality assurance (discussed in greater detail in Section 4) through training, extension support, and even acting as a guarantor of locally produced seed. This strong connection with the local/provincial government can be particularly important for building resilience in local seed systems, which has lessons for the current COVID-19 pandemic.

## 4. Results—Thematic Area: Liberalizing Seed Quality Control Mechanisms

### 4.1. Regulatory Aspects of Liberalizing Seed Quality Control Mechanisms

Seed quality control is a shared concern across countries' regulatory systems, and most countries maintain some regulatory conditions on the quality of seed that can be sold in the market. Regulatory approaches and flexibilities in this area vary. Flexibilities include systems that blend formal seed certification with more flexible models (including QDS, self-certification, and truth-in-labeling approaches) and those in which the certification process has been fully or partly privatized (including authorization of private seed inspectors). Figure 4 below highlights how the regulatory elements related to seed quality control impact farmers' abilities to acquire and sell seeds. Within this thematic area, the main regulatory element that applies is seed quality assurance mechanisms, although the scope of seed policy, law, and regulation can be relevant as well. Seed quality assurance mechanisms and the scope of rules and regulations affect, in particular, farmers' ability to acquire and exchange seeds of their choice; save, reuse, and exchange seed; and sell varieties locally and commercialize them more broadly. Regulatory flexibilities in this area include mixed systems for quality assurance, such as India's and Nepal's systems, that blend mandatory certification schemes (formal certification is not universally applied, but many governments seem to prefer this approach) with approaches that may open the door for a broader range of seeds in the market, such as quality declared seed schemes, standard seed classes, or truth-in-labeling approaches. Although fewer regulatory areas impact seed quality control than the other thematic area studied, regulatory flexibilities can be very important here as well, as examined in greater detail below.

### 4.2. Regulatory Approaches to Seed Quality Control

Seed quality control approaches can take several forms, and some countries even maintain mixed systems with several different regulatory approaches. A common approach, which tends to exhibit little flexibility unless combined with other mechanisms, is mandatory certification of all seed batches that are to be commercialized and sold. Formal seed certification can be time-consuming and costly, and it also requires an appropriate regulatory environment and quality control institutions, including a dedicated agency, sufficient certified laboratories, and qualified inspectors to perform inspections and certify seeds [49]. Formal seed certification is generally based on the Organisation for Economic Co-operation and Development (OECD) Seed Schemes and International Seed Testing Association (ISTA) standards, which smaller producers may find difficult to comply with. Because of the way formal certification systems are structured, they are also challenging to implement. In many cases, countries do not have sufficiently trained inspectors, appropriate laboratory equipment, or adequate capacity within the public sector to ensure a well-functioning formal seed certification system [46].

Several experts consulted in this study stressed that seed certification may not be the most pressing bottleneck for smaller farmers and could be approached collectively by informal sector stakeholders in order to successfully meet requirements. For example, seed clubs have helped farmers collectively certify seed [26]. However, formal seed certification

can still be costly and time-consuming, even with collective approaches. In Vietnam, for example, it took nine years for the HD1 rice variety to obtain formal certification [17].

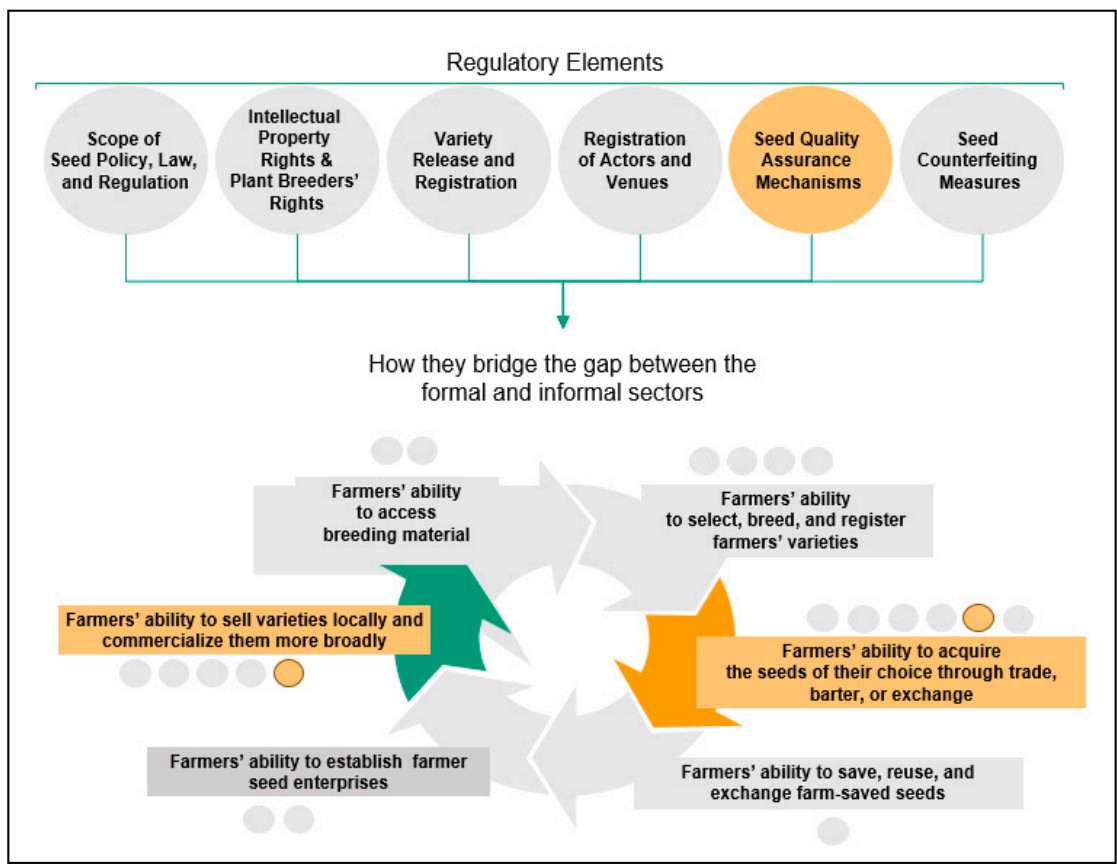

**Figure 4.** Liberalizing seed quality control mechanisms (adapted from [2,9]).

Formal seed certification also requires the involvement of the government, or a third party authorized by the government, to test and certify the quality of all seed lots. Several countries, such as Zambia, Kenya, Uganda, and Rwanda, allow for authorization of private seed inspectors to reduce the burden on public certifying agencies, and Tanzania and other countries are moving forward with such an approach as well [46]. Although generally considered a good regulatory practice, these public–private certification schemes have faced several implementation challenges, including, but not limited to, political resistance in the countries in which they have been adopted. Experts consulted have highlighted that these implementation challenges often correspond to resistance to changing systems that have been managed by, and relied upon, public interventions.

Other flexible practices have been developed to address quality control, such as QDS. Madagascar, Tanzania, and Uganda recognize QDS as an alternative method of quality assurance, which coexists alongside formal seed certification (see Table 2). The FAO has established QDS Guidelines to promote quality of seeds in regions where a formal seed certification system is difficult to implement or where varieties might not comply with the formal seed certification requirements [50]. Flexible practices in seed quality control vary from alternative quality assurance systems, such as QDS and truth-in-labeling, to mixed approaches that include the engagement of the private sector, including through authorization of private seed inspectors and the participation of seed clubs and associations.

**Table 2.** Key characteristics of seed quality control approaches.

| Seed Quality Control Approach | Country Examples | Key Characteristics |
|---|---|---|
| **Mandatory Certification** | **Zambia Kenya Uganda Rwanda Peru** | • Mandatory certification is contingent on variety and actors' registration (regulatory gateway/precondition).<br>• Formal seed certification is typically based on OECD Seed Schemes and ISTA Standards.<br>• Advanced regulatory environment and appropriate infrastructure needed.<br>• Involvement of public institutions or third-party seed certification bodies.<br>• Can incorporate private seed inspectors. |
| **Quality Declared Seeds (QDS)** | **Madagascar Tanzania Uganda Guatemala Peru Zambia** | • QDS can be used as a more flexible alternative for quality assurance than mandatory seed certification, especially for local and farmers' varieties.<br>• QDS can also be adopted to address inefficiencies and capacity gaps in formal seed certification mechanisms.<br>• QDS is often limited to specific regions or crops. |
| **Truth-in-Labeling** | **South Africa India Nepal** | • Truth-in-labeling relies on self-regulation of seed producers.<br>• Producers provide information based on minimum quality standards. |
| **Certification of Private or Third-Party Inspectors** | **Zambia Zimbabwe Kenya Peru** | • Certification of private inspectors requires government authorization and regulatory change.<br>• Private inspectors must meet certain requirements to operate. |
| **Seed Clubs and Associations** | **Vietnam Myanmar Zimbabwe** | • Seed clubs and associations can reduce costs and procedures for smallholder farmers. |

Many countries' regulations contain multiple "regulatory gateways or preconditions" and require that all seeds be certified and subject to requisite inspection before they can be commercialized; in addition, all farmers must be registered as seed producers, crop varieties must be formally registered, and, sometimes, the field in which a variety is produced must be registered. This is the case in Zambia, for example, and the Zambian Seed Control and Certification Institute (SCCI), oversees the certification process; however, SCCI can license any company or institution to conduct inspections, testing, and sampling of seed varieties. However, Zambia's system does contain flexibilities, and Zambia also has a somewhat "mixed" system, with QDS existing alongside formal seed certification as an alternative quality control mechanism for varieties that meet the minimum standards of germination and purity but cannot to comply with all the formal certification requirements.

In Vietnam, while the 2004 Seed Ordinance makes it illegal to sell non-certified seed, provincial authorities have allowed for some sale of small quantities of non-certified seed when limited to the community or district [17]. Seed quality evaluation is tasked to a unit within the club that comprises farmers who received training by the Seed Center through the CBCD-BUCAP (Community Biodiversity Development and Conservation—Biodiversity Use and Conservation in Asia Programme). The seed clubs have also worked with farmers to maintain quality standards; the seed clubs have seed inspectors who assess seed and label seed bags with the name of the variety and seal, if applicable, for certified varieties, or the name of the local variety or some identification codes for non-certified seeds [15]. For both certified and non-certified varieties, it is the Seed Center that approves the seed quality and the subsequent release of seeds into the market [17].

Authorization of private seed inspectors is a notable regulatory flexibility that is increasingly being considered by countries with formal seed certification systems. Zambia, Zimbabwe, and Kenya have all adopted this approach, which to some extent is modeled on the South African National Seed Organization (SANSOR) model. In these countries, a

public–private approach is followed that integrates accredited private seed inspectors into a formal seed certification system. South Africa's system, in contrast, follows a truth-in-labeling model and voluntary seed certification, except for certain crops, with SANSOR managing seed certification with a staff of 160 private seed inspectors.

Even with private seed inspection, however, the lack of infrastructure, including certified public laboratories, along with the fees for inspection and certification, can hinder farmers' access to formal seed certification. Peru's system also incorporates third-party certification (Box 4).

**Box 4.** Peru's third-party certification system.

Peru has implemented a seed certification system, whereby commercial seed can be certified through a third-party certification entity authorized by INIA, CODESEs (Comités Departamentales de Semillas, or Departmental Seed Committees). The CODESEs certify seeds of producers in the formal and informal seed sectors and also perform inspections, seed testing, and seed processing or storage for private enterprises. Third-party certification and seed certification activities are available in each of the CODESEs' eight seed service centers. This allows farmers to gain access to high-quality seeds across a wide variety of crops including rice, maize, potatoes, beans, and cotton (Iowa State University Seed Science Center 2011).

Peru has also established Technical Regulation for Seed Certification for seed certification and certification bodies (DS N° 0-24-2005-AG 2005). Peru's certifying body, SENASA (Servicio Nacional de Sanidad Agraria del Perú, or National Agrarian Health Service), ensures compliance of any natural or legal person seeking to conduct certification with the established requirements. In addition, Peru allows a form of QDS for traditional seeds.

In Vietnam, where seed clubs have played a central role in the seed sector, 30 percent of formal certified seed comes from smallholder farmers [26]. Similar models, such as seed cooperatives, have been successful elsewhere. In Zimbabwe, farmers have certified numerous varieties, including cowpea, sugar bean, maize, sorghum, and rice, through the ZAKA Super Seeds cooperative (Box 5).

**Box 5.** Zimbabwe's seeds cooperatives.

Smallholder farmers in Zimbabwe have struggled with limited land size, poor access to resources, and issues in the enabling environment. To address these gaps, the marginalized farmers of Zaka District in Zimbabwe have been supported through the Seeds and Markets Project (SAMP) funded by the Swiss Agency for Development Cooperation (SDC). The project focused on linking farmers to markets and increasing seed access and availability. Through the project, the farmers formed Zaka Seed Growers Association (Zaka Association) and also established a registered company, Zaka Super Seed, to manage the marketing of their seed. The Zaka Association was registered as a certifying authority in the 2010/2011 cropping season, which enabled it to produce certified seed of maize, sorghum, cowpea, rice and sugar beans and then market these varieties under its own brand name in later years. The project resulted in an increase in the area under seed production, with a shift in volume of seed produced by farmers in Zaka from 26 MT to 151 MT [51,52].

Another example of regulatory flexibility in systems with formal seed certification relates to how countries define seed classes. When narrowly defined, seed classes can be market limiting. However, broader seed classes exist, such as the class for standard grade seed in Zimbabwe (Uganda also includes standard seed in its seed classes). Zimbabwe recognizes four seed classes, Breeder Seed, Foundation Seed, Certified Seed, and Standard Grade Seed. Although, under Zimbabwe's Seed Regulations, some crops can only be commercialized if certified under the first three seed classes, other crops maybe commercialized as standard grade seed. The Crop Breeding Institute tests for germination and physical purity of these seeds before authorizing them for commercialization [53]. Zimbabwe's system shows how broader seed classes can provide an avenue for allowing different types of quality seed in the market.

While regional seed regulatory harmonization is not the focus of this study, harmonized seed classes are another aspect that warrants greater focus, given that regional rules

tend to standardize seed classes and sometimes do not provide for more flexible seed classes (such as standard grade seed) or alternative quality control measures, including QDS (except for SADC, as noted).

*4.3. Mixed Quality Control Systems*

In practice, many countries have mixed systems for quality control, which has helped to bridge the gap between the formal and informal sectors. In a mixed system, several quality control approaches exist alongside each other, with different rules and standards depending upon the approach. Mixed systems can include formal seed certification, QDS, truth-in-labeling, and self-certification. For example, many countries in Africa and Latin America maintain both a system for formal seed certification and QDS. Within Africa, examples include Tanzania, Zambia, and Uganda (Box 6).

**Box 6.** Case study—Uganda's QDS system.

> Uganda's rollout of QDS is particularly interesting, since it coexists alongside a formal certification system. Uganda's Seed Certification Service regulates the certification process under the Seed and Plant Act or 2006 and the 2017 Seed and Plant Regulations. While these regulations recognize six classes of seeds (breeder, pre- basic, basic, certified first generation, certified second generation, and standard seed), they also provide for use and sale of QDS as an alternative to formal certification. Uganda's QDS system incorporates many aspects of formal seed certification, including inspections, but reduces the number of inspections from six to at least one. Additionally, Uganda trains farmers' groups and associations to enable them to establish internal quality control mechanism that are then verified by the inspector [54].
> One of the benefits of this system is that QDS seed end up costing less than certified seed, which has proven to improve access to farmers, along with an increase in their revenues. This approach has established trust in the system similar to what formal certification would, while reducing costs and allowing farmers to invest resources in improvements in production. In Uganda, as is true elsewhere, QDS is limited to certain crops (self-pollinated and vegetatively propagated crops) and sales districts, which could be limitations to scaling the model.

Peru maintains both formal seed certification, with third-party inspection, and QDS. Peru's QDS system requires that seed producers be registered and take full responsibility for seed quality. Peru has adopted this parallel system to respond to the needs of farmers, whose seed varieties are mostly supplied by seed producers in the informal seed sector [8]. Under Peru's Seed Regulation, QDS seeds are classified as non-certified seed [8]. In Guatemala, QDS is formally recognized and has been used to introduce high-quality crops into remote areas to promote food security. The USAID Feed the Future project MAS-FRIJOL relied on QDS to increase bean production for smallholder farmers in Guatemala's highlands. Through MASFRIJOL, farmers from each community were trained in QDS for beans, which ultimately led them to produce a more nutritious crop of high-quality varieties and learn how to manage QDS systems [55].

According to some experts, QDS can be viewed as a relaxing of regulations rather than a positive application of standard new regulation [46]. In general, however, QDS is often intended to operate as an alternative quality control system that functions in parallel with formal seed certification, with government guidelines and minimum standards for certain crops through which farmers can guaranty the quality of their seeds [56]. Often, QDS is viewed as a bridge between the formal and informal seed sectors because it can be more accessible for small seed producers [2,54]. QDS has also been used to improve quality control of certain crop varieties, such as vegetatively propagated crops, that are as not as well recognized under formal quality control systems. While QDS is being used for an increasing range of crops, some experts consulted noted that QDS should remain limited only to certain crops.

QDS is often contained within certain territories or regions, meaning that QDS seed has a narrower reach in national markets than formally certified seed. Nevertheless, QDS can both provide a channel for farming communities to sell seed and can improve farmers'

access to quality seeds, which, in turn, can extend the seed market. It is important to note too that implementing QDS systems requires some infrastructure, including a system for adherence with guidelines and minimum quality standards, which can come at a cost. Use of QDS is not widespread, however, and there have been reports of skepticism on the part of farmers due to cost or value for use [9,49].

Experts consulted agreed that success in implementing alternative seed quality assurance mechanisms depends to a large extent on engagement with farmers' groups, cooperatives, and associations. For example, Laos and Uganda have relied upon a collective approach with interaction between the public sector and farmers. With the implementation of successful alternative mechanisms to seed quality control, certification becomes less of a pain point for farmers who can become more integrated into the formal seed market.

In some systems, quality assurance is left largely to seed producers under different forms of self-regulation. Truth-in-labeling requires producers to provide information related to the quality of the seed based on minimum standards (this will often appear on the package based on packaging and labeling rules), but it does not rest on mandatory certification [49]. As Box 7 highlights, India has adopted a mixed system that includes both a truth-in-labeling and voluntary certification approach.

**Box 7.** India's mixed system for seed quality control.

India's system allows for truthfully-labeled seed, along with voluntary certification of seed and some compulsory quality control [57]. Through this mixed system, all certified seed must be labeled, but all labeled seed need not be certified. This allows for a degree of self-regulation based on minimum standards, allowing small producers to only go through the certification system on a voluntary basis and thus reducing costs related to seed certification through a government agency [58]. Under India's formal seed certification system, the National Seeds Board accredits individuals and organizations who have fulfilled the prescribed seed certification procedures and affixed a certification tag/label. Similarly, for seed exports, certified seeds must meet the quality assurance requirements of the Seed Testing facilities, which have been established in conformity with ISTA and OECD seed standards. Truthfully-labeled seed, on the other hand, can be produced and sold within India by private entities with laboratory facilities and is priced lower than certified seed [59].

Nepal is another example of a mixed system. Nepal's Seed Rules admit a blend of seed quality control mechanisms, including both seed certification and truthful labeling for all seed classes (breeder seed, foundation seed, certified seed, and improved seed), as shown in Table 3 below. In Nepal, seed certification is voluntary and is carried out by authorized agencies; however, for seed that has not been certified, truthful labeling becomes a compulsory requirement and is carried out by seed producers. Notably, while quality assurance under seed certification is the responsibility of Nepal's certification agency, for truthful labeling, this responsibility lies with the producer.

**Table 3.** Comparison between seed certification and truthful labeling in Nepal [60].

| Element | Seed Certification | Truthful Labeling |
|---|---|---|
| **Type of seed** | Breeder seed, foundation seed, certified seed, improved seed | Breeder seed, source seed, label seed, improved seed |
| **Mandatory** | Voluntary | Compulsory if certification is not done. |
| **Who does the certification?** | Authorized agencies (Seed Quality Control Centre (SQCC) and Regional Seed Testing Laboratory) | Seed producers |
| **Procedure to follow** | Procedure set through regulations; inflexible | Flexible procedure; producers can allocate available time to monitor quality |
| **Who is responsible?** | Certification agency is responsible. | Producers are responsible |

*4.4. Key Findings Liberalizing Seed Quality Control*

Increasing regulatory flexibilities in seed quality control systems for informal seed actors can enhance farmers' ability to acquire the seeds of their choice and sell varieties locally. In some countries, mixed quality control systems allow for more formal seed certification to coexist alongside flexible regulatory approaches, including QDS and truth-in-labeling. To maintain quality of seed in the market while increasing regulatory flexibilities and reducing the number of regulatory gateways for informal seed actors, we found that:

- Alternative regulatory approaches and mixed regulatory systems, which combine different approaches to quality assurance, can mitigate challenges associated with formal seed certification systems, including the need for an advanced regulatory system and supporting infrastructure, heightened capacity of the public sector, and implementation gaps, among others.
- Formal seed certification processes are often viewed as cumbersome, costly, and inefficient in markets with large informal sectors and production of farmers', landrace, or traditional varieties. Alternative mechanisms for quality assurance exist and can be better adapted to farmers' varieties, including QDS and truth-in-labeling.
- While the benefits of QDS are still being assessed, Uganda's use of QDS shows that it can coexist alongside a formal certification system, reducing farmers' costs and increasing revenues [54].
- Public accreditation of private seed inspectors has been introduced in several countries (e.g., Peru, Zambia, Zimbabwe, and Kenya) to increase certification capacity. This is a promising model, although implementation challenges are still being reported in some systems.
- Truth-in-labeling, a quality assurance method that relies on the private sector, could be applied to a wider range of varieties if appropriate quality systems are established, including at the community level.
- Seed clubs, cooperatives, and community seed associations play a pivotal role in helping smaller farmers develop and maintain a commercial presence in the market while maintaining quality standards, as highlighted by the Vietnam and Zimbabwe case studies. In Vietnam, the local government worked with farmers to uphold standards set by the Ministry of Agriculture and Rural Development and provided essential training and extension support. This approach proved to be instrumental in both maintaining quality in the market and helping farmers bridge the divide between informal seed exchange and sale of formally certified rice varieties.
- Regulatory flexibility can also be applied through seed class definitions, and some governments have formally recognized both QDS and other seed classes such as "standard seed" that have paved the way for a wider range of quality seeds in the market. Currently, with S34D's support, Kenya is planning to pilot implementation protocols for standard seed certification approaches for a select set of crop varieties. At the regional level, SADC contains provisions recognizing QDS as a seed class, which is a notable innovation in regional trade law. Uganda and Zimbabwe have recognized "standard seed" and "standard grade seed" as seed classes, which allows for commercialization of a broader range of varieties.

## 5. Conclusions

This study presented a framework that evaluates how regulatory flexibility can be built into seed systems to engage farmers of all sizes. We focused on two study dimensions: (i) extending market frontiers (who can sell what seed, which crop varieties, where) and (ii) liberalizing seed quality control mechanisms (ranging from formal seed certification to quality declared seed (QDS) and self-certification).

Extending market frontiers directly impacts whether farmers can access seed of the right quality and variety at the right price to increase on-farm productivity, while seed quality control mechanisms affect the quality of seed available in the market. We find that flexible regulatory approaches and practices can build bridges between formal and

informal seed systems, guarantee quality seed in the market, and encourage market entry for high-quality traditional and farmer-preferred varieties.

There is not a "one size fits all model" for seed regulation, and, in some cases, variations in regulatory systems highlight important pathways for improving access, availability, and affordability of quality seed. National and local governments can adapt policy and regulatory options to local priorities, improving biodiversity and strengthening public–private partnerships in the process. As confirmed by the expert consultations conducted and case studies reviewed, regulatory flexibility can be built into even more formal, structured seed systems, creating space to expand the market frontier of a wider range of crop variety combinations and bridging gaps between formal and informal seed systems.

Resilience at the last mile is an important priority and is critical to absorb and adapt to shocks and pandemics. However, to build transformative capacity and expand choices for farmers at the last mile, a priority of S34D, farmers need access to a wider crop variety portfolio. This will depend upon practical solutions and applications of some of the regulatory flexibilities discussed in this paper that expand market frontiers by venues, actors, and crop varieties.

In some cases, more flexible local approaches, such as the local government's recognition of and support for the seed clubs in Vietnam, can be as efficient as formal approaches, particularly if both follow accepted standards and training and technical support are readily available to farmers. Notably, local approaches can be particularly resilient in the face of market disruptions, such as those caused by the current global pandemic, highlighting the need for strengthening local seed systems alongside national, formal systems.

Since it can sometimes be difficult to prioritize regulatory interventions, "regulatory gateways" that appear at different stages of the seed value chain and affect farmers' ability to participate in different activities can provide useful focal points. A regulatory gateway can take the form of a regulatory requirement to participate in the market (such as registration as a seed producer), link between one regulatory process and another (seed must be registered first and can only then be certified in order to be sold), or condition on market engagement (for example, the need to have a business address or meet a revenue threshold in order to be registered as a seed dealer). These gateways often highlight the most pressing obstacles for informal actors, and, as a result, they are some of the most useful intervention points for both policymakers and practitioners to bring about change in the regulatory environment for seed. Producers of some types of seed are better able to navigate formal seed systems (such as hybrid seed producers), while others may struggle within these systems. Regardless of the regulatory approach that a government chooses, revealed demand (or market pull) is a necessary condition for improved access and availability of seed, and application of legal and regulatory flexibilities will work best when the market demand for a variety exists.

In addition to these observations, we find that:

- Designing interventions to improve seed regulatory systems requires assessing and understanding the national context, including the system of national law (common law, civil law, mixed approach, etc.), the interplay between the national and sub-national government, and the relative level of development of the various seed systems in the country.
- Seed laws and regulations must be considered when designing any business models that impact seed systems in a country. This will impact long-term viability and sustainability in the market and factor into the "demand pull" needed to create long-term market development.
- Regulatory gateways link one part of the seed system to another, and these linkages (e.g., the link between seed variety registration and commercial sale) can be very important intervention points for strengthening both informal and formal seed systems. In some cases, regulatory gateways can be reduced by establishing different requirements for different types of seed in the context of the same regulatory process,

such as Malaysia's identifiability test for crop variety registration and plant breeder's rights for farmers', community, and traditional varieties.

- Regulatory practices at different market stages (e.g., pre-commercialization and commercialization) can also highlight important common ground. For example, the way in which QDS has developed to allow for more local commercialization of seed (see Section 4 on quality control mechanisms) could provide insight for building flexibility into crop variety registration. QDS seeds sometimes do not need to undergo formal variety registration in order to be sold in the market, but they are still subject to quality control measures before they can be commercially sold, highlighting that some regulatory considerations can be prioritized while others are treated more flexibly under certain circumstances. It is, however, important to note that QDS can also come with challenges, including limitations in crops and market and the higher costs associated with decentralized monitoring and quality assurance [49].
- Country ownership is an essential component of successfully introducing regulatory flexibilities or reducing regulatory gateways. In particular, the roles of local, national, and regional governments will be critical, as the case studies cited have shown. In essence, capacity building is not just about seed companies, but it also involves farmers, non-governmental bodies, and government stakeholders.
- As some examples in our study have highlighted, given the right opportunities and platforms, farmers can develop and release their own varieties, which can then be scaled up for wider market distribution.
- Local seed systems need to be strong to sustain shocks and build resilience. In designing and implementing policies, decision-makers must factor in the role that informal seed systems play in last-mile markets.

This study highlights promising trends and interesting variations in regulatory approaches. While we saw several examples of flexibility in regulatory design, we found sparse evidence on implementation, which is largely anecdotal at this stage. Going forward, we hope the study can foster South–South learning to gather additional information on best practices and implementation approaches. Similarly, as a community, we need to monitor, evaluate, and learn from the usefulness and impact of flexible regulatory approaches that establish a stronger connection between informal and formal seed systems.

**Author Contributions:** Conceptualization, B.D. (study thematic areas) and K.K. (regulatory flexibilities and approach); methodology, K.K. (regulatory methodology and approach, regulatory flexibilities and gateways, legal and regulatory literature review and analysis) and B.D. (stakeholder prioritization, seed systems literature review); software (N/A); validation, K.K. and B.D.; formal analysis, K.K.; investigation, K.K. and B.D.; resources, Catholic Relief Services/USAID (see disclaimer below); data curation (N/A); writing—original draft preparation, K.K.; writing—review and editing, B.D. (abstract, introduction, conclusion) and K.K.; visualization, New Markets Lab; supervision (N/A); project administration, B.D.; funding acquisition, Catholic Relief Services/USAID (see disclaimer below). All authors have read and agreed to the published version of the manuscript.

**Funding:** This study was conducted under the Feed the Future Global Supporting Seed Systems for Development (S34D) activity implemented by Catholic Relief Services. This activity focuses on bridging gaps between seed systems and is funded by the U.S. Government's Feed the Future Initiative through the Bureau for Resilience and Food Security (RFS) and by the United States Agency for International Development (USAID) through the Bureau for Humanitarian Assistance (HA). Disclaimer: This work was funded in whole (or part) by the United States Agency for International Development (USAID) under Agreement 7200AA18LE00004 as part of Feed the Future Supporting Seed Systems for Development. The views expressed herein are the author's own and do not represent the views of USAID or the U.S. Government.

**Institutional Review Board Statement:** Not applicable.

**Informed Consent Statement:** Informed consent was obtained from all subjects involved in the study, and key stakeholders interviewed were part of a webinar where the study results were disseminated.

**Data Availability Statement:** The study did not report any data.

**Acknowledgments:** The authors wish to thank A.A., D.B., I.B., C.C., B.d.J., N.L., S.M., D.O., G.O., R.T., M.S., and B.V. for their insights and contributions provided in the development of this study. Expert comments provided in the development of this study are not attributed, unless approved and noted. The authors also wish to thank A.M.G.E., M.F.A., N.K., and F.O. for their extensive research support.

**Conflicts of Interest:** The authors declare no conflict of interest.

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
