# Peer review of "Using Regulatory Flexibility to Address Market Informality in Seed Systems: A Global Study"

_agronomy, doi:10.3390/agronomy11020377_

Round 1

Reviewer 1 Report

the Paper is very timely especially during this period of glonal poliy discussions around integrated seed systems. the paper is also very well written and well thought out. it is an interesting read with good case studies.

My few comments are attached.

Author Response

NOTE:  PDF includes edits to respond to all comments received, which we greatly appreciated.  

  1. The paper is generally very well written and though out. It is also timely as global policy discussions around seed systems are on-going in many aspects at different international for a such as Governing Body meeting of the IT, Commission for genetic resources for food and agriculture etc. there are still few comments as outlined below.
  2. The title should reflect the contents of the paper. i.e “Using Regulatory Flexibility to Address Market Informality in Seed Systems: A Global Study” ADDRESSED -- Title changed to "Using Regulatory Flexibility to Address Market Informality in Seed Systems:  A Global Study"
  3. The paper could also do with much more discussion on the interface between the formal and the informal seed systems especially the recent global developments on integrated seed systems including what is driving this system i.e the demand side related to farmers’ needs and the fact that farmers in many developing countries grow different “types” of seeds to meet different needs e.g . ADDRESSED Lines 67-102
  4. The paper could also do with a very good (and broad) definition of seed because to capture the interphase between “seed and genetic resources” and how each are treated differently by different regulatory systems i.e. different sets of policies may dictate/affect the way “seed” or “genetic resources” are used for instance ADDRESSED Lines 74-80
    1. According to the International Treaty for plant genetic resources for food and agriculture, Seeds/genetic resources in farmers’ hands are not considered to be in the multilateral system of access and benefit sharing and hence their access by breeders etc are subject to negotiations under ABS of Nagoya protocol. Their access, breeding and eventual registration and release may follow a different path in terms of intellectual property rights etc Is this addressed sufficiently? Addressed at Line 553
    2. Accessing genetic resources for breeding could eventually affect the way the seed is released, the IPRs placed on the seed and benefits shared and although this may not directly affect the quality control system, it could still affect the value chain pertaining to variety release and registration and hence availability Addressed at Line 551
  5. The seed regulatory value chain is interesting however it could do with the addition of seed dissemination (after breeding) and (exports) in between somewhere– this is also in line with the argument about how RECs harmonization e.g. in the case of COMESA and SADC would affect how seed quality assurance is done and how marketing is done to fulfil obligations related to the agreements or to access certain markets. All these eventually affect access of the seed to farmers (cost and availability) ADDRESSED Line 207 to include dissemination and explain that labeling, packaging (including seeds allowed in small packs), marketing, and distribution (including the free distribution by institutional buyers).  Sentence also added that regional harmonization applies to the entire value chain Addressed at Line 208
  6. I like figure 4 very much and would be nice if it is linked to article 9 of the treaty on farmers’ rights which is already mentioned in the paper Addressed beginning at Line 287

The conclusion can also discuss the “cons” of some flexible quality control arrangements eg QDS ( https://edepot.wur.nl/251357) especially in relation to the cost of “decentralizing seed inspectors or the costs of monitoring QDS seeds of farmers  Addressed at Line 926

Reviewer 2 Report

The topic of the papers is very interesting.

Authors should better explain what determines the existence of a formal or informal seed system. Is the market size? Does the possibility or convenience of patenting a type of seed play a role in creating a formal or informal seed sector? These are probably relevant elements in creating a bridge between the two seed systems.

The explanation of figures 1,2,3, and 4 should be added in the text.

Author Response

NOTE:  PDF includes edits to respond to all comments received, which we greatly appreciated.  

The topic of the papers is very interesting.

Authors should better explain what determines the existence of a formal or informal seed system. Is the market size? Does the possibility or convenience of patenting a type of seed play a role in creating a formal or informal seed sector? These are probably relevant elements in creating a bridge between the two seed systems. ADDRESSED Lines 67-102

The explanation of figures 1,2,3, and 4 should be added in the text. ADDRESSED 

Reviewer 3 Report

This article surveys the different seed rules and regulations that affect how actors are regulated in the breeding, registration, production, purchase, transportation and sale of seeds and how the quality of seed are regulated. This is an incredibly important study, as there has been limited comprehensive information on the different types of regulatory systems in seed systems and how to create synergies and bridge actors and varieties across the informal and formal seed system. In addition, this article is well written and comprehensive.  A few minor revisions.

  • The title of the paper does not reflect the main subject of the paper: seed systems. I almost passed up the review because I thought I knew nothing about the topic. Please add something pertaining to seed or seed systems in the title. It really isn’t enough that it is in the keywords.
  • On Page 2 the authors indicate that informal systems are largerly unregulated and dependent upon farmer knowledge. Indeed, they are also dependent on trust and social capital. If someone you know shares or sells you poor quality seed, then you just lost a friend and or business. Seed from informal systems in my experience (perhaps dependent upon crop) are well regulated in situations where they are shared or sold locally within known social networks. This is important to point out, and if necessary, to define what you mean by regulated. There are different forms of “regulation” that fall outside of what we may consider “formal” regulation. I suggest acknowledging or indicating evidence for what has been fairly effective “informal” regulation of the informal seed sector.
  • Did I miss the discussion of theme 3 – Improving anti-counterfeiting? I expected there to be a section on this theme just as there was Theme 1 and 2. The paper stands alone without it, or with a smaller section addressing counterfeiting. But it is confusing in the introduction that there are 3 themes and only in-depth discussion of the first two. Please revise/address this issue.
  • It seems like footnote 2 is important to the methods section and understanding the authors’ approach to the work. Please consider including in materials and methods, or explain why it isn’t there.
  • In material and methods – how did you go about identifying the different countries you end up profiling or drawing examples from? There are obviously a lot of countries that could be used as examples for having differentiated or mixed, or non-differentiated markets and it is a little confusing as one gets deeper into the paper how the authors decided to focus on the countries that they did. These countries are obviously interesting case studies, but more on how you narrowed down, or how many country regulatory systems you looked at, would be nice to know. I would like to see an explanation. Even if it was narrow in some sense, this does not take away from the study. It simply needs an explanation.
  • The figures need more explanation. I sent an inordinate amount of time trying to understand what they are showing and the differences between the regulatory ones.
    • Figure 1, I suppose the repetitive gray bubbles line up with the value chain steps, but then the blue ones kind of float out in nowhere and the slight variation in wording in them is not elucidating. In fact, what are the circles? 5 of the gray ones say the same thing and it is not completely clear how they link with the other parts of the figure. The value chain part of this figure is useful for the reader and aligns well with your point that regulation can and cannot happen all along the value chain, the rest of it is confusing as it stands.
    • Figure 2 and 3. Again, an explanation is needed. Why is it a darker turquoise line linking some of the upper circles? The “how they bridge the gap between the formal and informal sectors” seems like it should read, how these regulatory elements influence farmer’s ability to…It isn’t completely clear how the circles bridge the gap.
    • Figure 3 – two circles are greyed out. What does this mean? Please include in figure caption.
    • Where is Benin in the boxes with country summaries?
    • Again, how did you come to focus in on these countries?
    • You mention the Vietnam seed clubs early on, with no elaboration and the reader is remains wondering what it is about these seed clubs that demonstrate the point you are making.
    • The boxes explaining specific systems are great.
  • What are some examples of countries where there is no flexibility?
  • 317 VCU – deserves a footnote as to what it is. DUS seems to be more generally understood.
  • 324 – 326: Traditional or landraces were not bred/developed/ or maintained by farmers with the intention for DUS. These varieties have variability for resilience built into them purposefully and regulatory systems that do not value this may actually be detrimental to the integrity of these varieties/landraces. You indicate farmer knowledge, but it isn’t clear how you mean this. I suggest the authors elaborate on it (perhaps the authors are trying to acknowledge that farmers intentionally have variability in their varieties). It is important to acknowledge this in addition to your point.
  • 335 The authors indicate that ‘Malaysia applies an “identifiability” approach to assess varieties based on combination of qualities’ What is the identifiability approach? This needs a little bit more explanation.
  • Line 332-4 This wording is confusing. Please revise.
  • 378 consider including a subsection here for provincial and regional regulations. It needs a bit of a transition and subsection title would suffice.
  • In the beginning of section 4.2, the authors discuss “one end of the regulatory spectrum” – I kept looking for the other end of the spectrum. This is explained a few paragraphs down, but it might be clearer if you indicate that when you start to talk about it. Just a suggestion.
  • In the same section, an explanation of the value of certification might be useful. It is clear that it is challenging to implement, so why do we care about it? Is in necessary everywhere?
  • The final comment I have is on a larger concept that is not touched on in this paper. Why does the informal seed system need regulation? In the intro the authors indicate that the types of regulations can either link the formal and informal or further cause separation. Obviously, there can be benefit to increased access to improved varieties when (IF) they are suitable to the last mile. Once the informal system is regulated, does this limit other forms of access? Does it open opportunities for misguided use of the regulations? Does it enable power to remain in the hands of those that understand regulations and can enforce regulations? The cases the authors bring up show the value of differentiated and mixed systems, but regulations may impose a specific world view and process.

Author Response

NOTE:  PDF includes edits to respond to all comments received, which we greatly appreciated.  

The title of the paper does not reflect the main subject of the paper: seed systems. I almost passed up the review because I thought I knew nothing about the topic. Please add something pertaining to seed or seed systems in the title. It really isn’t enough that it is in the keywords. -- Title changed to "Using Regulatory Flexibility to Address Market Informality in Seed Systems:  A Global Study"

On Page 2 the authors indicate that informal systems are largerly unregulated and dependent upon farmer knowledge. Indeed, they are also dependent on trust and social capital. ADDRESSED at Line 96

If someone you know shares or sells you poor quality seed, then you just lost a friend and or business. Seed from informal systems in my experience (perhaps dependent upon crop) are well regulated in situations where they are shared or sold locally within known social networks. This is important to point out, and if necessary, to define what you mean by regulated. There are different forms of “regulation” that fall outside of what we may consider “formal” regulation. I suggest acknowledging or indicating evidence for what has been fairly effective “informal” regulation of the informal seed sector. ADDRESSED at Line 96

Did I miss the discussion of theme 3 – Improving anti-counterfeiting? I expected there to be a section on this theme just as there was Theme 1 and 2. The paper stands alone without it, or with a smaller section addressing counterfeiting. But it is confusing in the introduction that there are 3 themes and only in-depth discussion of the first two. Please revise/address this issue. ADDRESSED and removed (we had covered three themes in an older version but pared this back to the two thematic areas)

It seems like footnote 2 is important to the methods section and understanding the authors’ approach to the work. Please consider including in materials and methods, or explain why it isn’t there. ADDRESSED and pulled into text starting at Line 71

In material and methods – how did you go about identifying the different countries you end up profiling or drawing examples from? There are obviously a lot of countries that could be used as examples for having differentiated or mixed, or non-differentiated markets and it is a little confusing as one gets deeper into the paper how the authors decided to focus on the countries that they did. These countries are obviously interesting case studies, but more on how you narrowed down, or how many country regulatory systems you looked at, would be nice to know. I would like to see an explanation. Even if it was narrow in some sense, this does not take away from the study. It simply needs an explanation. ADDRESSED at Line 213

The figures need more explanation. I sent an inordinate amount of time trying to understand what they are showing and the differences between the regulatory ones. ADDRESED throughout

    • Figure 1, I suppose the repetitive gray bubbles line up with the value chain steps, but then the blue ones kind of float out in nowhere and the slight variation in wording in them is not elucidating. In fact, what are the circles? 5 of the gray ones say the same thing and it is not completely clear how they link with the other parts of the figure. The value chain part of this figure is useful for the reader and aligns well with your point that regulation can and cannot happen all along the value chain, the rest of it is confusing as it stands. ADDED explanations to Figures
    • Figure 2 and 3. Again, an explanation is needed. Why is it a darker turquoise line linking some of the upper circles? The “how they bridge the gap between the formal and informal sectors” seems like it should read, how these regulatory elements influence farmer’s ability to…It isn’t completely clear how the circles bridge the gap. ADDRESSED Added explanations to Figures
    • Figure 3 – two circles are greyed out. What does this mean? Please include in figure caption. Added explanations to Figures
    • Where is Benin in the boxes with country summaries? ADDRESSED by noting that examples in box are illustrative
    • Again, how did you come to focus in on these countries? ADDRESSED at Line 213
    • You mention the Vietnam seed clubs early on, with no elaboration and the reader is remains wondering what it is about these seed clubs that demonstrate the point you are making. ADDRESSED at Lines 139, 271, 306 (Vietnam Seed Clubs also discussed in Box 2)
    • The boxes explaining specific systems are great. Thank you!
  • What are some examples of countries where there is no flexibility? Most countries show some degree of flexibility, which was the focus of our assessment.  We are uncomfortable citing examples of countries with no flexibility, since this would require a comprehensive review of a number of legal instruments, which was outside of our scope.
  • 317 VCU – deserves a footnote as to what it is. DUS seems to be more generally understood. ADDED reference to national performance trials to help explain at Line 443 and 450
  • 324 – 326: Traditional or landraces were not bred/developed/ or maintained by farmers with the intention for DUS. These varieties have variability for resilience built into them purposefully and regulatory systems that do not value this may actually be detrimental to the integrity of these varieties/landraces. You indicate farmer knowledge, but it isn’t clear how you mean this. I suggest the authors elaborate on it (perhaps the authors are trying to acknowledge that farmers intentionally have variability in their varieties). It is important to acknowledge this in addition to your point. ADDED at Line 456
  • 335 The authors indicate that ‘Malaysia applies an “identifiability” approach to assess varieties based on combination of qualities’ What is the identifiability approach? This needs a little bit more explanation. ADDRESSED by pulling out of footnote and into text at Line 469
  • Line 332-4 This wording is confusing. Please revise. ADDRESSED
  • 378 consider including a subsection here for provincial and regional regulations. It needs a bit of a transition and subsection title would suffice. COMMENT:  We did not cover provincial regulation in a comprehensive manner, so we did not wish to add a section or sub-section; however, there is a reference at Line 521 and Box 2 on Vietnam
  • In the beginning of section 4.2, the authors discuss “one end of the regulatory spectrum” – I kept looking for the other end of the spectrum. This is explained a few paragraphs down, but it might be clearer if you indicate that when you start to talk about it. Just a suggestion. ADDRESSED
  • In the same section, an explanation of the value of certification might be useful. It is clear that it is challenging to implement, so why do we care about it? Is in necessary everywhere? ADDED short reference to government preference for certification as a quality assurance system at Line 660
  • The final comment I have is on a larger concept that is not touched on in this paper. Why does the informal seed system need regulation? In the intro the authors indicate that the types of regulations can either link the formal and informal or further cause separation. Obviously, there can be benefit to increased access to improved varieties when (IF) they are suitable to the last mile. Once the informal system is regulated, does this limit other forms of access? Does it open opportunities for misguided use of the regulations? Does it enable power to remain in the hands of those that understand regulations and can enforce regulations? The cases the authors bring up show the value of differentiated and mixed systems, but regulations may impose a specific world view and process. ADDRESSED in Abstract and Line 154
